

# Effects of different resistance training frequencies on body composition and muscular performance adaptations in men

Hamid Arazi[1], Abbas Asadi[2], Paulo Gentil[3], Rodrigo Ramírez-Campillo[4,7], Pooria Jahangiri[1], Adel Ghorbani[1], Anthony C. Hackney[5] and Hassane Zouhal[6]

[1] Department of Exercise Physiology, Faculty of Sport Sciences, University of Guilan, Rasht, Guilan, Iran
[2] Department of Physical Education and Sport Sciences, Payame Noor University, Rasht, Guilan, Iran
[3] Faculdade de Educação Física e Dança, Universidade Federal de Goiás, Goias, Brazil
[4] Department of Physical Activity Sciences, Universidad de Los Lagos, Osorno, Chile
[5] Department of Exercise & Sport Science; Department of Nutrition, University of North Carolina, Chapel Hill, North Carolina, United States
[6] M2S (Laboratoire Mouvement, Sport, Santé) – EA 1274, Univ Rennes, Rennes, France
[7] Centro de Investigación en Fisiología del Ejercicio, Facultad de Ciencias, Universidad Mayor, Santiago, Chile

Corresponding author
Hamid Arazi,
hamid.arazi@guilan.ac.ir

## ABSTRACT

**Background:** The aim of this study was to compare the effects of 8 weeks resistance training (RT) with two sessions versus four sessions per week under volume load-equated conditions on body composition, maximal strength, and explosive actions performance in recreationally trained men.

**Methods:** Thirty-five healthy young men participated in the study and were randomly divided into a two sessions per-week RT (RT2, $n = 12$), four sessions per-week RT (RT4, $n = 13$) or a control group (CG, $n = 10$). All subjects were evaluated for thigh, chest and arm circumference, countermovement jump (CMJ), medicine ball throw (MBT), 1-repetition maximum (1RM) leg press, bench press, arm curl, muscular endurance (i.e., 60% of 1RM to failure) for leg press, and bench press at pre, mid (week 4) and post an 8-week training intervention.

**Results:** A two-way analysis of variance with repeated measures (3 [group] × 3 [time]) revealed that both training groups increased chest and thigh circumferences, strength and explosive actions performance tests in comparison to CG following 8 weeks of training ($p = 0.01$ to $0.04$). Group × time interactions were also noted in 1RM bench press (effects size [ES] = 1.07 vs. 0.89) and arm curl (ES = 1.15 vs. 0.89), with greater gains for RT4 than RT2 ($p = 0.03$).

**Conclusion:** RT improved muscle strength, explosive actions performance and markers of muscle size in recreationally trained men; however, four sessions of resistance training per week produced greater gains in muscular strength for the upper body measures (i.e., 1RM bench press and arm curl) when compared to two sessions per week under volume-equated conditions.

## INTRODUCTION

Resistance training (RT) is an exercise modality commonly used to improve muscle hypertrophy and strength (*ACSM, 2009*; *Fleck & Kraemer, 2004*). Designing an optimum RT program requires controlling variables such as the number of sets, repetitions, intensity, exercise selection–sequence, and rest intervals (*Fleck & Kraemer, 2004*). Recently, some studies have focused on the effects of RT frequency on muscular adaptations (*Arazi & Asadi, 2011*; *Dankel et al., 2017*; *Saric et al., 2018*; *Gentil et al., 2015*). The frequency of RT describe the number of training sessions performed per muscle group in a given period (*ACSM, 2009*), which is commonly restricted to a week (*Dankel et al., 2017*).

Previous studies have typically compared 1 vs. 2, 1 vs. 3, 3 vs. 4, and 3 vs. 6 times per week RT frequencies on muscular adaptations, with controversial findings (*Arazi & Asadi, 2011*; *Dankel et al., 2017*; *Saric et al., 2018*; *Gentil et al., 2015, 2018*; *Brigatto et al., 2018*; *Colquhoun et al., 2018*; *Gomes et al., 2018*; *Häkkinen & Kallinen, 1994*; *Raastad et al., 2012*; *Zaroni et al., 2019*; *Schoenfeld et al., 2015*; *Yue et al., 2018*). For example, when *Colquhoun et al. (2018)* and *Saric et al. (2018)* compared 3 vs. 6 days per week RT on muscular adaptations in resistance-trained men, with volume equated, both frequencies induced similar gains in strength and muscle hypertrophy. In addition, *Brigatto et al. (2018)* concluded that both one and two RT session per week promoted neuromuscular adaptations including muscular strength and endurance with a similar change between experimental conditions. Similarly, other authors reported similar changes in muscle strength and hypertrophy with equal volume RT performed one or two times per week in untrained (*Gentil et al., 2015*) and trained men (*Gentil et al., 2018*). In contrast, *Zaroni et al. (2019)* examined well-trained men, with a split training routine with muscle groups trained once per week vs. whole-body split training routine with muscle groups trained 5 days per week, and found that higher frequencies induced superior hypertrophic effect. Moreover, in a series of systematic review studies by *Schoenfeld, Ogborn & Krieger (2016)* and *Schoenfeld, Grgic & Krieger (2018)* the authors addressed that twice weekly RT in more effective than once a week RT to increase muscle hypertrophy.

The controversy between studies may derive from previous limitations among published studies. For example, when RT programs of different frequency are performed under volume-equated conditions, muscle strength gain is similar between different frequencies (*ACSM, 2009*; *Schoenfeld, Ogborn & Krieger, 2016*; *Schoenfeld, Grgic & Krieger, 2018*). Another caveat in the literature is that comparisons are usually limited to muscle strength and hypertrophy (*Saric et al., 2018*; *Brigatto et al., 2018*; *Gomes et al., 2018*; *Zaroni et al., 2019*; *Schoenfeld et al., 2015*), and little is known about the effects of RT frequencies on muscle power and endurance performance in recreationally trained individuals. Moreover, randomized-controlled interventions, with an equated volume load between different training frequencies are lacking. Therefore, the purpose of this study was to investigate the effects of volume load-equated RT frequencies of 2 vs. 4 times per week on muscular strength, endurance, power performances, and muscle size in recreationally trained young men.

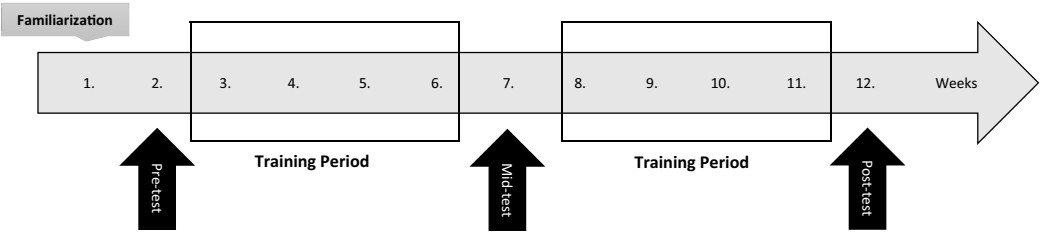

**Figure 1 Study design.**

# METHODS

## Study design

In a randomized-controlled longitudinal design, subjects were divided into 3 groups, including RT performed 2 times per week (RT2), 4 times per week (RT4) and a control group (CG). The study duration lasted 12 weeks (Fig. 1). The main training intervention period lasted 8 weeks and the subjects performed equal volume training with differing training frequencies (i.e., 2 vs. 4 times per week). Pre, mid and post 8-week training, one repetition maximum (1RM) of leg press, bench press, and arm curl, muscular endurance (i.e., 60% of 1RM to failure) for the upper- and lower-body (i.e., bench press and leg press), countermovement jump and medicine ball throw, in addition to thigh, chest and arm circumferences were measured. Two measurements with 96 h apart were used to determine the reliability of tests and the intraclass correlation coefficient (ICC) of all tests were r ≥ 0.95.

## Participants

Thirty-five young men who recreationally trained RT (i.e., 2 or 3 days per week for at least 2 years) participated in this study. Inclusion criterions for the study were (1) no upper- and lower-body injuries or orthopedic problems as screened by physician, (2) no medical problems or any history of ankle, knee, or back pathology in the 3 months before the study, (3) no lower or upper-body reconstructive surgery of any type in the past 2 years or unresolved musculoskeletal disorders, (4) no problems of the cardiovascular and endocrine systems. Furthermore, the subjects were required to not have used any supplement or drug within the past 6 months prior to inclusion in this study which was confirmed by a personal interview. The subjects were assigned to 3 groups including: 2 times per week RT (RT2; $n$ = 12, age = 19.8 ± 1.8 y, height = 1.75 ± 0.5 m, mass = 64.2 ± 5.7 kg, body fat = 16.6 ± 4.9%, and training age = 2.5 ± 0.5 y), 4 times per week RT (RT4; $n$ = 13, age = 19.9 ± 1.6 y, height = 1.77 ± 0.4 m, mass = 70.6 ± 8.2 kg, body fat = 18.0 ± 4.1%, and training age = 2.8 ± 0.7 y) and a control group (CG; $n$ = 10, age = 20.4 ± 1.4 y, height = 1.78 ± 0.8 m, mass = 69.1 ± 8.0 kg, body fat = 18.4 ± 3.7%, and training age = 2.3 ± 0.4 y) using computer-generated random numbers (Fig. 2). After being informed about the study procedures, benefits and possible risks, the participants signed an informed consent form in accordance with the guidelines of the Institutional Review Board at the University of Guilan (Project. 1398/2019).

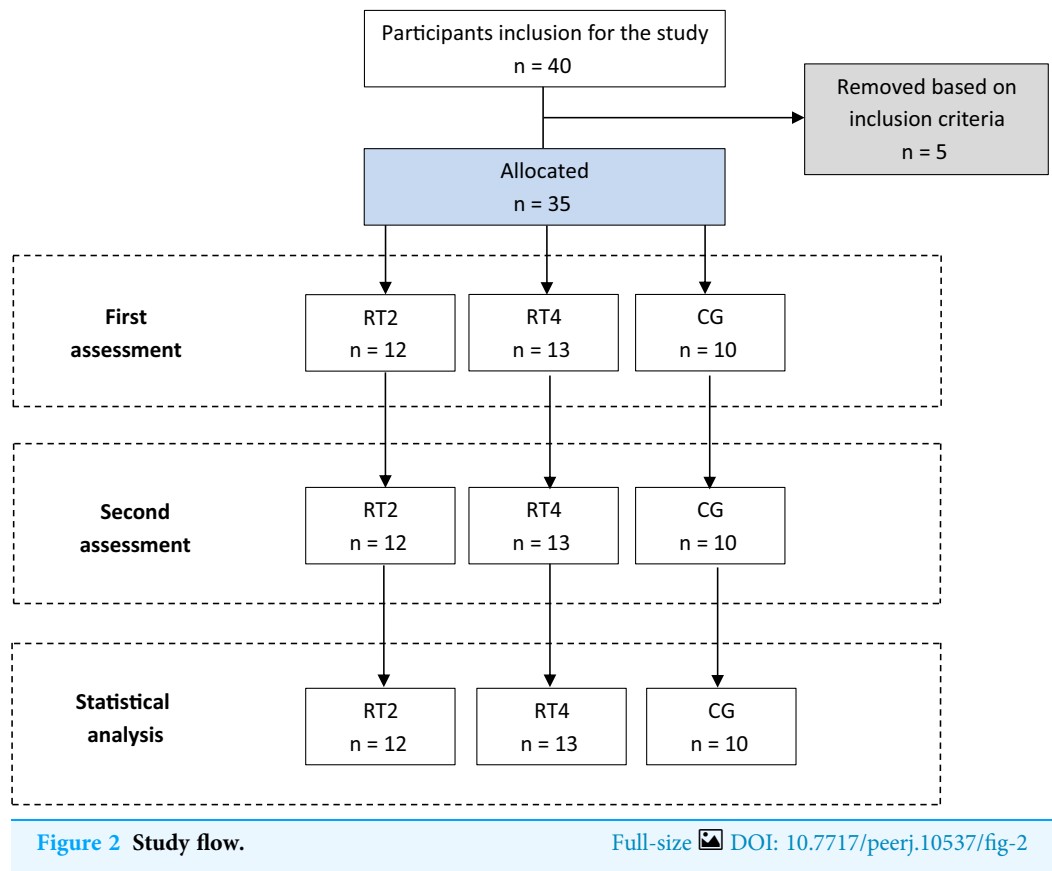

**Figure 2 Study flow.**

## Procedures

The volunteers visited the laboratory 9 times for testing including 3 days for pre-test (24 h apart between testing sessions), 3 days for mid-test (24 h apart between testing sessions), and 3 days for post-test (24 h apart between testing sessions). The subjects were tested at the same time of day (4 to 6 P.M.) and in the same order to minimize the effect of circadian variations in the test results. All subjects were instructed to continue with their normal daily life activities and dietary intake throughout the study duration.

## Anthropometric measures

Height was measured using a wall-mounted stadiometer (Seca 222, Terre Haute, IN, USA), body mass was measured using a medical scale (BC-418MA; Tanita, Tokyo, Japan), and skinfold thickness was measured at 3 sites (i.e., pectoral, quadriceps, and abdominal) on the right side of the body using calipers (model 01128; Lafayette Caliper, Lafayette, IN, USA) (*Jackson & Pollock, 1985*). Each site measurement was assessed 3 times and the average of 3 trials was recorded for analysis. The circumferences of chest, mid-thigh, and mid-arm on the right side of the body were assessed using tape measure with nearest to 0.1 cm (*Arazi, Damirchi & Asadi, 2013*). The arm and thigh circumferences were measured with the muscle maximally contracted. All anthropometric measures were assessed by the same researcher who was experienced and qualified for the measurements.
**Table 1 Dietary intake assessed for the RT2, RT4 and control groups at pre and in the middle of the training period (mean ± SD).**

|  |  | RT2 | RT4 | Control |
|---|---|---|---|---|
| Energy intake (kcal) | Pre | 2,632 ± 310 | 2,521 ± 276 | 2,618 ± 288 |
|  | Mid | 2,991 ± 298 | 2,892 ± 199 | 2,632 ± 299 |
| Carbohydrate (g) | Pre | 270 ± 33 | 269 ± 39 | 272 ± 37 |
|  | Mid | 292 ± 41 | 298 ± 41 | 288 ± 33 |
| Fat (g) | Pre | 85 ± 21 | 87 ± 23 | 81 ± 33 |
|  | Mid | 94 ± 24 | 92 ± 21 | 82 ± 38 |
| Protein (g) | Pre | 108 ± 22 | 105 ± 21 | 100 ± 19 |
|  | Mid | 129 ± 26 | 125 ± 30 | 98 ± 22 |
| Vitamin E (mg) | Pre | 9.6 ± 1.0 | 9.3 ± 1.2 | 9.1 ± 1.1 |
|  | Mid | 11.0 ± 1.5 | 10.7 ± 1.4 | 9.0 ± 0.8 |
| Vitamin C (mg) | Pre | 72 ± 18 | 71 ± 17 | 70 ± 15 |
|  | Mid | 79 ± 21 | 78 ± 13 | 71 ± 18 |

## Diet control

To avoid potential dietary confounding of results, 3-day diet recalls were completed at pre- and mid study duration, and the subjects were advised to maintain their customary nutritional regimen (i.e., approximately 25% protein, 25% fat and 50% carbohydrate) and to avoid taking any supplements during the study period. The nutrition specialist continued to meet with the subjects each week to assess adherence to their food and liquid instructions and avoidance of drugs and ergogenic supplements using interview before the initiation of each training session (Table 1).

## Muscular strength

Lower body muscular strength was assessed with the leg press exercise, upper-body muscular strength was assessed using the free-weight barbell bench press and arm curl exercises, respectively. The one repetition maximum (1RM) testing was performed according to method previously described in detail (*Arazi, Damirchi & Asadi, 2013*; *Fleck & Kraemer, 2004*). Briefly, the subjects performed a warm-up set of 8 to 10 repetitions at a light weight (∼50% of 1RM). A second warm-up consisted a set of three to five repetitions with a moderate weight (∼75% of 1RM), and third warm-up included one to three repetitions with a heavy weight (∼90% of 1RM). After the warm-up, each subject was tested for the 1RM by increasing the load during consecutive trials until the subjects were unable to perform a proper lift, complete the range of motion, and/or maintain correct technique. The 1RM test was determined by ∼5 sets of one repetition, with 3–5 min of rest among attempts.

A bilateral leg press test was selected to provide data on maximal dynamic strength through the full range of motion of the muscles involved. Bilateral leg press tests were completed using standard a 45° leg press machine (Nebula Fitness, Inc., Versailles, OH, USA), with the subjects assuming a sitting position (about 120° flexion at the hips, 80° flexion at the knees, and 10° dorsiflexion) and the weight sliding obliquely at 45°.

A manual goniometer (Q-TEC Electronic Co. Ltd., Gyeonggi-do, South Korea) was used at the knee to standardize the range of motion. On command, the subjects performed a concentric leg extension (as fast as possible) starting from the flexed position (85°) to reach the full extension of 180° against the resistance determined by the weight. The free-weight barbell (DHZ Barbell Model, Tehran, Iran) bench press is a valid and specific method to assess upper-body strength performance. This test initiated with the arms fully extended, holding the weight directly above the chest. The weight is lowered at a controlled speed and with a smooth motion, to just touch the chest then returned to the starting position. The free-weight barbell (DHZ Barbell Model, Tehran, Iran) arm curl is used as a valid method to assess hand muscle strength. This test initiated in standing position holding barbell using two hands with the arms hanging by the side of body. The elbows were in extending position and then the elbows are closed up to shoulder level while contracting the biceps muscle. The spotters and an experienced strength and conditioning coach provided verbal encouragement and ensure safety.

## Muscular endurance

Before the endurance test, the subjects performed a short period of warm-up including 5 min of running and 5 min of stretching exercise and then performed 10 repetitions with 30–40% of 1RM for each exercise test. The muscular endurance tests were performed according to method previously described in detail (*Arazi, Asadi & Roohi, 2014*). Briefly, after warm-up, the subjects performed as many repetitions as possible without stopping or pausing between repetitions with 60% of 1RM to exhaustion with 1 h rest between the two tests (i.e., bench press and leg press) (*Arazi & Asadi, 2011*).

## Lower and upper body power performance

Lower body power performance was measured at first, using the countermovement jump test (CMJ). For the CMJ, subjects performed standard warm-up including 10 min light running and ballistic movements and then performed five CMJs without arms akimbo with 30-s rest period (*Arazi, Asadi & Roohi, 2014*). The Vertec (Muscle LabV718; Ergo Jump Plus Bosco System, Langesund, Norway) was adjusted to match the height of the individual participant by having him stand with the dominant side to the base of the testing device. The dominant hand was raised and the Vertec was adjusted so that the hand was the appropriate distance away from the marker based on markings on the device itself. The subjects were instructed to flex their knees until 90° according to previously established methods (*Arazi, Asadi & Roohi, 2014*). Each subject performed 3 maximal CMJ with 30-s rest period and the greatest jump recorded for further analysis.

Upper body power performance was measured 30 min post CMJ test, using the medicine ball throw (MBT). For the MBT, subjects performed standard warm-up including 10 min of light stretching and ballistic movements for the upper body and then performed five balls throwing with 30-s rest period. The subjects sat on the floor and flexed their elbow similar to basketball chest pass and push the ball (3 kg Rubber Medicine Ball; Champion Sports, Taiwan) as far as possible. There was a floor-mounted tape measure that was used to record distance from sitting position to first contact of the ball.

**Table 2 Resistance training protocol.**

| RT2 group | Saturday | Repetitions | Tuesday | Repetitions |
|---|---|---|---|---|
| | Leg press | 10-10-8-8 | Leg extension | 10-10-8-8 |
| | Lying leg curl | 10-10-8-8 | Deadlift | 10-10-8-8 |
| | Lat pull down | 10-10-8-8 | Lat rowing | 10-10-8-8 |
| | Bench press | 10-10-8-8 | Incline bench press | 10-10-8-8 |
| | Lateral raises | 10-10-8-8 | Military press | 10-10-8-8 |
| | Machine biceps curl | 10-10-8-8 | Arm curl | 10-10-8-8 |
| | Machine triceps extension | 10-10-8-8 | Lying triceps extension | 10-10-8-8 |
| **RT4 group** | **Saturday and Tuesday** | **Repetitions** | **Sunday and Wednesday** | **Repetitions** |
| | Leg press | 10-8 | Leg extension | 10-8 |
| | Lying leg curl | 10-8 | Deadlift | 10-8 |
| | Lat pull down | 10-8 | Lat rowing | 10-8 |
| | Bench press | 10-8 | Incline bench press | 10-8 |
| | Lateral raises | 10-8 | Military press | 10-8 |
| | Machine biceps curl | 10-8 | Arm curl | 10-8 |
| | Machine triceps extension | 10-8 | Lying triceps extension | 10-8 |

Note:
RT2: 2 times per week resistance training, RT4: 4 times per week resistance training, RM: repetition maximum.

In fact, the distance of the throw of the medicine ball from sat position up to its first contact with the ground was measured as upper body power. Each subject performed five maximal MBT with 30-s rest period (Abe et al., 2000) and the greatest distance recorded for further analysis.

## Resistance training program

Table 2 presented the summary of the RT program. The training protocol included a mixture of single-joint and multi-joint exercises with equated training volume load (repetitions × external load [kg]) between experimental groups. A 60 to 90 s period of rest between sets and 2 to 3 min of rest between exercises were allowed. The RT intensity was between 70% to 80% of 1RM which determined by 1RM testing prior to inclusion in study schedule and weight was increased systematically if the prescribed amount of repetitions were completed. Each training session was supervised by a researcher and Certificated Strength and Conditioning Specialist, with a coach: trainee ratio of 1:5 (Gentil & Bottaro, 2013). To continuously provide appropriate loading based on the current strength levels of the subjects, they tested at pre-training and after 4 weeks of training to modify RT intensity.

## Statistical analyses

A two-way analysis of variance with repeated measures (3 [group] × 3 [time]) was used to determine significant differences among groups. Assumptions of sphericity were assessed using Mauchly's test of sphericity, with any violations adjusted by use of the Greenhouse-Geisser (GG) correction. When a significant F value was achieved, Bonferroni post hoc procedures were performed to identify the pairwise differences between the

means. Customized excel spread sheets were used to calculate all effect size (ES) statistics. Hedge's g (g = (Mpost – Mpre)/SDpooled) was utilized to calculate an effect size for all measures. Threshold values for assessing magnitudes of ES were <0.2, trivial; 0.2–0.6, small; 0.6–1.2, moderate; 1.2–2.0, large; 2.0–4.0, very large; and >4.0, nearly perfect (*Hopkins, Marshall & Batterham, 2009*). The effect size is reported with the 95% confidence interval (CI) for all analysed measures. All data are presented as mean ± SD. The ICC was used to determine the reliability of the measurements. The level of significance was set at $p \leq 0.05$. The statistical tests were performed using the SPSS statistical package version 21 (Chicago, IL, USA).

## RESULTS

The test–retest reliability coefficient of all variable tests was r ≥ 0.95. At baseline, no significant differences were observed among groups in any dependent variables ($p = 0.642$). In addition, the CG did not show significant changes at any time point in the variables ($p = 0.211$).

There was no significant difference between the RT2 (week 4 = 45.37 ± 5.62 kg, week 8 = 91.37 ± 11.51 kg) and RT4 (week 4 = 48.68 ± 6.77 kg, week 8 = 93.28 ± 12.42 kg) in the training volume load at week 4 ($p = 0.52$) and week 8 ($p = 0.46$).

There were significant time effects which indicated significant increases in chest and thigh circumferences at mid and post-training intervention for both the RT2 and RT4 ($p = 0.01$). No significant increase was seen in the arm circumference for both the groups ($p = 0.6$). There were significant group by time interaction in chest ($p = 0.018$) and thigh ($p = 0.026$) circumference increases following 8 weeks of training which indicated significant differences between trained groups than CG at mid and post-test values. However, no significant differences were observed between RT2 and RT4 in chest and thigh circumferences at mid and post-test (Tables 3 and 4).

There were time effects which indicated significant increases in 1RM of bench press, leg press and arm curl at mid and post-training intervention for both the RT2 and RT4 ($p = 0.001$). There were group by time interaction in 1RM of bench press ($p = 0.031$) and arm curl ($p = 0.022$) following 8 weeks of training which indicated statistically significant differences between the RT4 compared with RT2 at post-test. Compared with CG, both the RT2 and RT4 groups indicated significant differences at mid- and post-test ($p = 0.001$) in all strength measures (Tables 3 and 4).

There was a time effect which indicated significant increases in leg press endurance at mid and post-training intervention for both the RT2 and RT4 ($p = 0.001$). There was a significant group by time interaction ($p = 0.041$) in leg press endurance which indicated significant increases between the trained groups than the CG at mid and post-test values. However, no significant differences were observed between RT2 and RT4 in leg press endurance at mid and post-test (Tables 2 and 3). In bench press endurance, there was a time effect which indicated significant increases at mid and post-training intervention for the RT4 ($p = 0.001$). There was a significant group by time interaction ($p = 0.032$) in bench press endurance which indicated significant differences between the RT4 than the CG at mid and post-test values (Tables 2 and 3). However, no significant

**Table 3 Changes in anthropometric and performance variables in response to 8 weeks training intervention (mean ± SD).**

| Variable | Group | Testing time | | | Statistics |
|---|---|---|---|---|---|
| | | Pre | Mid | Post | |
| Chest circumference (cm) | RT2 | 86.4 ± 4 | 88.7 ± 4.3* | 89.8 ± 4.5*‡ | G = 0.08 |
| | RT4 | 86.5 ± 7.3 | 91.3 ± 8.6*‡ | 92.5 ± 8.1*‡ | T = 0.001 |
| | CG | 86.3 ± 6.8 | 87.1 ± 7.5 | 86.8 ± 7.2 | G × T = 0.018 |
| Thigh circumference (cm) | RT2 | 53.6 ± 3.3 | 56.0 ± 3.7*‡ | 56.1 ± 3.4*‡ | G = 0.54 |
| | RT4 | 54.7 ± 7.6 | 56.2 ± 6.0*‡ | 56.3 ± 6.3*‡ | T = 0.001 |
| | CG | 53.5 ± 4.4 | 52.1 ± 3.4 | 51.8 ± 4.2 | G × T = 0.026 |
| Arm circumference (cm) | RT2 | 27.2 ± 2.5 | 28.2 ± 2.9 | 28.8 ± 2.5 | G = 0.1 |
| | RT4 | 29.3 ± 3.4 | 29.8 ± 3.4 | 29.5 ± 3.1 | T = 0.6 |
| | CG | 27.6 ± 2.1 | 27.0 ± 1.2 | 26.6 ± 1.2 | G × T = 0.12 |
| 1RM bench press (kg) | RT2 | 63.5 ± 6.4 | 70.3 ± 7.3*‡ | 71.5 ± 10.5*‡ | G = 0.12 |
| | RT4 | 64.6 ± 5.3 | 69.8 ± 10.6*‡ | 75.5 ± 12.8*†‡** | T = 0.001 |
| | CG | 64.4 ± 7.5 | 63.5 ± 7.5 | 64.8 ± 6.2 | G × T = 0.031 |
| 1RM leg press (kg) | RT2 | 201.2 ± 36.6 | 260.6 ± 48.0*‡ | 310.3 ± 52.5*†‡ | G = 0.48 |
| | RT4 | 203.4 ± 51.7 | 263.9 ± 68.5*‡ | 299.8 ± 64.2*†‡ | T = 0.01 |
| | CG | 202.8 ± 45.2 | 204.1 ± 39.8 | 203.4 ± 41.2 | G × T = 0.47 |
| 1RM arm curl (kg) | RT2 | 28.7 ± 4.0 | 33.6 ± 4.0*‡ | 34.3 ± 8.2*‡ | G = 0.1 |
| | RT4 | 28.6 ± 5.3 | 35.8 ± 6.2*‡ | 37.2 ± 8.7*‡** | T = 0.001 |
| | CG | 29.1 ± 3.1 | 29.1 ± 2.5 | 29.9 ± 3.1 | G × T = 0.022 |
| Bench press endurance (repetitions) | RT2 | 21.1 ± 3.6 | 21.5 ± 3.6 | 21.5 ± 3.8 | G = 0.11 |
| | RT4 | 21.5 ± 2.6 | 23.0 ± 3.1*‡ | 23.5 ± 2.3*‡ | T = 0.01 |
| | CG | 19.4 ± 4.4 | 19.8 ± 4.6 | 20.1 ± 3.5 | G × T = 0.032 |
| Leg press endurance (repetitions) | RT2 | 20.7 ± 7.0 | 24.1 ± 4.2*‡ | 26.5 ± 5.4*‡ | G = 0.07 |
| | RT4 | 19.4 ± 4.8 | 27.0 ± 4.9*‡ | 27.2 ± 6.4*‡ | T = 0.001 |
| | CG | 20.2 ± 3.3 | 20.8 ± 5.5 | 19.3 ± 4.2 | G × T = 0.041 |
| Countermovement jump (cm) | RT2 | 37.3 ± 4.4 | 41.7 ± 5.2*‡ | 43.7 ± 3.3*‡ | G = 0.36 |
| | RT4 | 37.8 ± 4.6 | 41.6 ± 3.4*‡ | 42.8 ± 5.5*‡ | T = 0.021 |
| | CG | 37.4 ± 3.7 | 37.0 ± 3.9 | 37.1 ± 4.6 | G × T = 0.047 |
| Medicine ball throw (m) | RT2 | 3.49 ± 0.52 | 3.64 ± 0.45*‡ | 3.72 ± 0.37*‡ | G = 0.87 |
| | RT4 | 3.72 ± 0.5 | 3.86 ± 0.58*‡ | 3.99 ± 0.65*‡ | T = 0.029 |
| | CG | 3.49 ± 0.35 | 3.48 ± 0.4 | 3.49 ± 0.51 | G × T = 0.048 |

Notes:
* Significant differences compared to pre-value.
† Significant differences compared to mid-value.
‡ Significant differences compared to CG.
** Significant differences between training groups.
RT2: 2 times per week resistance training, RT4: 4 times per week resistance training, CG: control group.
G: group, T: time.

differences were observed between RT2 and RT4 in bench press endurance at mid and post-test (Tables 2 and 3).

There were time effects which indicated significant increases in CMJ and MBT at mid and post-training intervention for both the RT2 and RT4 ($p = 0.02$). There were significant

Table 4 Time point ES in anthropometric and performance variables in response to 8 weeks training intervention.

| Variable | Group | ES (95% Cl) | | |
|---|---|---|---|---|
| | | Pre to mid | Mid to post | Pre to post |
| Chest circumference (cm) | RT2 | 0.53 [−0.28 to 1.35][b] | 0.24 [−0.56 to 1.04][b] | 0.77 [−0.06 to 1.6][c] |
| | RT4 | 0.58 [−0.2 to 1.37][b] | 0.14 [−0.63 to 0.91][a] | 0.75 [−0.04 to 1.55][c] |
| | CG | 0.11 [−0.77 to 0.98] | −0.04 [−0.92 to 0.84] | 0.07 [−0.81 to 0.95] |
| Thigh circumference (cm) | RT2 | 0.66 [−0.13 to 1.45][c] | 0.03 [−0.74 to 0.8][a] | 0.72 [−0.07 to 1.52][c] |
| | RT4 | 0.21 [−0.59 to 1.01][b] | 0.02 [−0.78 to 0.82][a] | 0.22 [−0.58 to 1.02][b] |
| | CG | −0.22 [−1.1 to 0.65] | −0.08 [−0.95 to 0.8] | −0.38 [−1.26 to 0.51] |
| Arm circumference (cm) | RT2 | 0.36 [−0.42 to 1.13][b] | 0.21 [−0.56 to 0.99][b] | 0.62 [−0.17 to 1.41][c] |
| | RT4 | 0.14 [−0.66 to 0.94][a] | 0.09 [−0.71 to 0.89][a] | 0.06 [−0.74 to 0.86][a] |
| | CG | −0.34 [−1.22 to 0.55] | −0.32 [−1.2 to 0.56] | −0.56 [−1.45 to 0.33] |
| 1RM bench press (kg) | RT2 | 0.96 [0−.15 to 1.77][c] | 0.13 [−0.64 to 0.9][a] | 0.89 [−0.08 to 1.7][c] |
| | RT4 | 0.6 [−0.22 to 1.42][b] | 0.47 [−0.34 to 1.28][b] | 1.07 [−0.22 to 1.93][c] |
| | CG | −0.11 [−0.99 to 0.76] | 0.18 [−0.7 to 1.06] | 0.06 [−0.82 to 0.93] |
| 1RM leg press (kg) | RT2 | 1.35 [0.5 to 2.2][d] | 0.96 [0.15 to 1.77][c] | 2.33 [1.34 to 3.33][e] |
| | RT4 | 0.96 [0.12 to 1.81][c] | 0.52 [0.29 to 1.34][b] | 1.6 [0.68 to 2.52][d] |
| | CG | 0.03 [−0.85 to 0.91] | −0.02 [−0.89 to 0.86] | 0.01 [−0.86 to 0.89] |
| 1RM arm curl (kg) | RT2 | 1.17 [0.22 to 2.12][c] | 0.11 [−0.66 to 0.87][a] | 0.84 [−0.04 to 1.64][c] |
| | RT4 | 1.21 [0.34 to 2.08][d] | 0.18 [−0.62 to 0.98][a] | 1.15 [0.29 to 2.02][c] |
| | CG | 0.0 [−0.88 to 0.88] | 0.27 [−0.61 to 1.15] | 0.25 [−0.63 to 1.13] |
| Bench press endurance (repetitions) | RT2 | 0.11 [−0.66 to 0.88][a] | 0.0 [−0.77 to 0.77][a] | 0.1 [−0.66 to 0.87][a] |
| | RT4 | 0.51 [−0.31 to 1.32][b] | 0.18 [−0.62 to 0.98][a] | 0.79 [−0.04 to 1.62][c] |
| | CG | 0.09 [−0.79 to 0.96] | 0.07 [−0.81 to 0.95] | 0.17 [−0.71 to 1.05] |
| Leg press endurance (repetitions) | RT2 | 0.57 [−0.21 to 1.35][b] | 0.48 [−0.3 to 1.26][b] | 0.9 [−0.09 to 1.71][c] |
| | RT4 | 1.51 [0.61 to 2.42][d] | 0.03 [−0.77 to 0.83][a] | 1.33 [0.45 to 2.22][d] |
| | CG | 0.13 [−0.75 to 1] | −0.29 [−1.17 to 0.59] | −0.23 [−1.11 to 0.65] |
| Countermovement jump (cm) | RT2 | 0.88 [0.08 to 1.69][c] | 0.44 [−0.33 to 1.22][b] | 1.59 [0.71 to 2.48][d] |
| | RT4 | 0.91 [0.07 to 1.75][c] | 0.25 [−0.55 to 1.06][b] | 0.95 [0.11 to 1.8][c] |
| | CG | 0.1 [−0.78 to 0.98] | 0.02 [−0.85 to 0.9] | 0.07 [−0.81 to 0.95] |
| Medicine ball throw (m) | RT2 | 0.3 [−0.47 to 1.07][b] | 0.19 [−0.58 to 0.96][a] | 0.49 [−0.29 to 1.27][b] |
| | RT4 | 0.25 [−0.55 to 1.05][b] | 0.2 [−0.6 to 1.01][a] | 0.45 [−0.36 to 1.26][b] |
| | CG | 0.03 [−0.85 to 0.9] | 0.0 [−0.88 to 0.88] | 0.02 [−0.85 to 0.9] |

Notes:
[a] Trivial.
[b] Small.
[c] Moderate.
[d] Large.
[e] Very large ES.
RT2: 2 times per week resistance training, RT4: 4 times per week resistance training, CG: control group.

group by time interaction ($p = 0.04$) in CMJ and MBT which indicated significant differences between the trained groups than the CG at mid and post-test values (Tables 2 and 3). However, no significant differences were observed between the RT2 and RT4 in CMJ and MBT at mid and post-test (Tables 3 and 4).

## DISCUSSION

The aim of the present study was to examine the effects of an 8-week RT program performed two or four times per week RT with equal weekly training volume on thigh, arm, and chest circumferences, 1RM of back squat, bench press, and arm curl, muscular endurance and explosive actions performance for the upper- and lower-body in recreationally trained young men.

In circumference measures, both the training groups significantly increased from pre-to-post RT intervention in the chest and thigh circumferences, without significant change for the arm circumference. In addition, the gains in this marker of muscle size were similar between the RT2 and RT4 groups (small to moderate ES, Table 3), with the exception of pre-to-mid and pre- to-post, where the RT2 group that indicated moderate ES while the RT4 group indicated small ES without statistically significant differences. The findings of the present study are in accordance with other studies that have reported improvements in muscle size after RT with varied training frequencies (*Arazi & Asadi, 2011*; *Saric et al., 2018*; *Colquhoun et al., 2018*; *Schoenfeld, Ogborn & Krieger, 2016*; *Schoenfeld, Grgic & Krieger, 2018*). In relation to the effects of training frequency on changes in muscle size or muscular hypertrophy, *Schoenfeld, Ogborn & Krieger (2016*; *Schoenfeld, Grgic & Krieger, 2018)* and *Grgic, Schoenfeld & Latella (2018)* reported small (i.e., range between ES = 0.22 to 0.51) gains using different RT frequencies, while in this study we found moderate (0.75 to 0.77 ES) gains in chest circumference after both RT2 and RT4. Previous experimental studies reported that RT interventions with two sessions per week induced small gains (i.e., 0.33 ES) but in this study we found moderate (range between 0.62 to 0.77 ES) increases in arm, thigh and chest muscle size. This suggest that RT with a frequency of at least 2 days per week is adequate to enhance muscle size (*Gentil et al., 2015*; *Colquhoun et al., 2018*; *Zaroni et al., 2019*; *Yue et al., 2018*). The RT2 group performed 4 sets per exercise in each training session, which may induce stimulation of muscular hypertrophy, by signalling pathways that increase protein synthesis and providing mechanical stress in the muscle fibers (*Fernandes et al., 2012 Padilha et al., 2019*). However, it seems that the muscle hypertrophy expansion is more impressed by volume of training and, considering that both groups trained at what has been shown to be the optimal dose (*Barbalho et al., 2019*), it can be derived that frequency of RT might play a subsidiary figure relevant to this and further investigations are needed to illuminate the effects of training frequency under volume-equated conditions on muscle size. In addition, whilst circumference measures has been shown to be reliable and reproducible and might be an appropriate field-centred criterion (*De França et al., 2015*), to make careful deductions based on the evidence, subsequent studies should focus on the use of direct gauges of muscle mass increase using MRI, DXA, ultrasound or BIA; however, previous studies used the aforementioned equipment and reported small gains in muscle hypertrophy using different training frequencies (*Gentil et al., 2015*, *Colquhoun et al., 2018*; *Zaroni et al., 2019*; *Yue et al., 2018*; *Schoenfeld, Ogborn & Krieger, 2016*; *Schoenfeld, Grgic & Krieger, 2018*).

Both RT groups increased their 1RM after 4 and 8 weeks training intervention. To date, a large number of studies reported that RT is an optimum training modality for strength enhancement in men and women (*Abe et al., 2000*; *Arazi, Damirchi & Asadi, 2013*; *Arazi, Asadi & Roohi, 2014*). In relation to strength gains following the first 4 weeks of training, aside from muscular hypertrophy, neuromuscular changes may have taken place (i.e., inter-muscular consonance ameliorations, augmented alpha motor-neurons firing rate, modified mechanical specifications of the muscle-tendon complex, ordonnance and/or individual-fiber mechanics) (*Loenneke et al., 2019*).

The RT4 group gained significantly greater strength than the RT2 in 1RM of bench press and arm curl following 8 weeks of training. However, with comparing ES the RT2 indicated large and very large changes in 1RM of leg press following 4 and 8 weeks training intervention. *Grgic et al. (2018)* in the review article addressed that muscular strength is increased due to more training frequencies; however, RT frequency does not show meaningful effect on muscular strength improvements while equated training volume. They reported moderate ES for 2 and 4 times per week RT frequency (i.e., 0.83 and 1.08, respectively), whereas we found similar gains in bench press and arm curl but large and very large ES in the 1RM leg press. The possible discrepancy in results could be due to type of test measures such as multi-joint vs. single joint (i.e., leg press vs. knee extension) and upper vs. lower body tests. Another possible mechanism for the greater strength gains in bench press and arm curl 1RM after the RT4 compared to the RT2 could be due to motor learning viewpoint. In fact, multi-joint motions including more mixed RT exercises need to an accurate coordination and timing of muscle recruitment and a greater grade of motor efficiency (*Carroll, Riek & Carson, 2001*). Therefore, increases in RT frequency from 3 to 4 sessions per week would provide more exposure to a given test/exercise, which can lead to a higher performance on that test (*Mattocks et al., 2017*) and hence resulted in greater upper body strength gains in the RT4 group.

Our findings demonstrated significant changes in lower and upper-extremity muscular endurance for the RT2 and RT4 groups after 4 and 8 weeks training intervention. These results are according to the last studies that displayed improvements in muscular endurance following RT (*Aagaard et al., 2002*; *Arazi, Damirchi & Asadi, 2013*). When comparing the ES, the RT4 showed more gains than RT2 in the endurance tests belonging to leg press and bench press (Table 3). The possible explanation for these findings could be due to the greater possibility of higher frequencies to enhance cellular adaptations (i.e., mitochondrial biogenesis) to increase muscle endurance; however, the information respective to this issue is rare and further studies are needed to explain the influence of different RT frequencies on muscular endurance performance.

Both RT groups increased their upper- and-lower body power performance after 4 and 8 weeks training intervention. In line with the results of this study, previous studies reported improvements in power performance after RT (*Arazi & Asadi, 2011*; *Arazi, Asadi & Roohi, 2014*). Typically, increases in CMJ and MBT performance following first 4 weeks of training and continually to 8 weeks training intervention could induced by neuromuscular adaptations (*Aagaard et al., 2002*). In fact, an improvement in power performance in the early stages of a strength training program is likely the result of

adaptations in the nervous system (*Assunção et al., 2016*). In fact, *Aagaard et al. (2002)* reported that the principal components of the training enforced progressions following RT were elucidated by elevations in efferent neural drive. This may be one explanation for the changes in lower- and upper-body power performance (i.e., CMJ and MBT) after the RT intervention. However, this is the first study that compared the effects of an 8 week RT with either 2 or 4 weekly training sessions on upper and lower-body power performance. The distribution of training volume on either 2 or 4 weekly training sessions yielded similar effects on power performance for stretch-shortening cycle tasks in CMJ and MBT tests. The observed performance enhancements could be explained by inter-muscular coordination improvements, increased alpha motor-neurons firing rate, improved mechanical characteristics of the muscle-tendon complex, improved muscle size, architecture and/or single-fiber mechanics (*Arazi & Asadi, 2011*; *Arazi, Asadi & Roohi, 2014*); however, more studies are needed to clarify the impact of training frequencies on power related performance adaptation following RT.

## CONCLUSION

RT improved muscle strength, power performance and markers of muscle size in recreationally trained men; however, four sessions of resistance training per week produced greater gains in muscular strength when compared to two session per week under volume load-equated conditions. Two and four times per week RT induced significant effects on muscular adaptations following 8 weeks of training in recreationally trained young men. In addition, RT for 4 times per week induced further adaptive responses in muscular strength in bench press and arm curl. It can be recommended that strength and conditioning professionals keep in their mind that 4 times a week RT could be adequate for muscular strength gains and 2 times a week RT could be suitable for the muscle size and power performance under volume load-equated conditions.

## ACKNOWLEDGEMENTS

The authors are thankful to the participants of this study for their excellent collaborations.

### Funding

The authors received no funding for this work.

### Competing Interests

Rodrigo Ramirez Campillo is an Academic Editor for PeerJ.

### Author Contributions

- Hamid Arazi conceived and designed the experiments, analyzed the data, authored or reviewed drafts of the paper, and approved the final draft.
- Abbas Asadi conceived and designed the experiments, analyzed the data, prepared figures and/or tables, authored or reviewed drafts of the paper, and approved the final draft.

- Paulo Gentil analyzed the data, authored or reviewed drafts of the paper, and approved the final draft.
- Rodrigo Ramírez-Campillo analyzed the data, authored or reviewed drafts of the paper, and approved the final draft.
- Pooria Jahangiri performed the experiments, prepared figures and/or tables, and approved the final draft.
- Adel Ghorbani performed the experiments, prepared figures and/or tables, and approved the final draft.
- Anthony C. Hackney analyzed the data, authored or reviewed drafts of the paper, and approved the final draft.
- Hassane Zouhal analyzed the data, authored or reviewed drafts of the paper, and approved the final draft.

### Human Ethics

The following information was supplied relating to ethical approvals (i.e., approving body and any reference numbers):

The Institutional Review Board at the University of Guilan approved this study (Project.1398/2019).

### Data Availability

The raw measurements are available in the Supplemental Files.

### Supplemental Information

Supplemental information for this article can be found online at http://dx.doi.org/10.7717/peerj.10537#supplemental-information.

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
