# Peer review of "Effects of different resistance training frequencies on body composition and muscular performance adaptations in men"

_PeerJ, doi:10.7717/peerj.10537_

## Round 0.1 · original submission · Major Revisions

While there was considerable variation in the three reviewers comment, I feel there is sufficient merit in this research study to warrant a resubmission opportunity. Please take on board all the three reviewers comments, especially reviewer two in your resubmission. In particular, you need to satisfy this reviewer that your participants are experienced in resistance training. I agree with their interpretation that the baseline absolute 1RM scores (particularly for the bench press) are relatively low for men with any resistance training experience. However, while I may have missed it, I didn't see any reports of their height and body mass. Therefore, I think you need to report their baseline body mass as it is still possible that these 1RM scores may be close to body weight, which does support some degree of resistance training experience.

Reviewer 1 ·

Basic reporting

The manuscript was written in a clear manner that was easy to follow, and professionally presented throughout.

Several suggestions that may improve the quality of the manuscript follow:

In the abstract:
- It may be useful to include ES's for the changes in 1RM bench press and arm curl for the 2x and 4x groups so the magnitude of this significant change is clear (lines 42-44).
- Ensure you are clear that the four sessions produced a greater gain in strength for the upper body measures only (lines 45-46).

In the methods:
- Change "In randomized-controlled..." to "In a randomized-controlled..." (line 91).
- Add a comma after "n = XX" is stated for both the 2 times group (line 105) and 4 times group (line 107)
- "instructed" may be a better word that "oriented" (line 122).
- Be consistent in use of a space (or no space) between numbers and "%" symbols (e.g. line 148 vs line 151).
- Ensure brackets are included around the Mpost - Mpre in the hedges g formula (line 181).

In the results:
- Change "...statistically significant increases for..." to "...statistically significant differences between..." in lines 202-203.
- Change "...significantly increases..." to "...significant increases..." in lines 199, 206, 212, and 217.

In the discussion:
- Change "show" to "shown" in line 251.
- Change "...occurred than muscular hypertrophy" to "...occurred, not only muscular hypertrophy" (line 265)
- Saying "more exposure to" may be more appropriate than "more time" in line 279.

In the conclusion:
- "adequate" may be a better word choice than "suitable" in line 309.

In the tables:
- In Table 2, I would suggest a different word than "significant" for the final column, perhaps "statistics"
- In Table 3, use superscript for a, c, b, and d, in your notes when defining the key used.

In the figures:
- In Figure 2, it may be useful to briefly state the reasons for the five subjects being removed.

Experimental design

Overall the experiment was well designed, including the two experimental groups as well as a control group. I also liked the use of dietary control, ensuring the participants were educated and kept on track here - an often overlooked (or under controlled) aspect of training studies.

I have several queries/suggestions for this section:
- The ICC of tests is stated as r ≥ 0.95 (lines 99-100), was this for all tests? Please state this, or for which tests specifically.
- The 2 times per week group initially weighed 64.2 ±5.7 kg and the 4 times per week group initially weighed 70.6 ±8.2 kg (lines 106-108); were the groups checked for differences at baseline?
- How was the height measured for the MBT? (line 159-163).
- Where the RM training prescriptions initially prescribed as percentages? Or were these RM's tested prior to the study? (line 168 and Table 1).

Validity of the findings

- I would be cautious in your statements in lines 243-245. You found no significant differences between groups for muscle size, thus I don't think you can make this assumption, regarding 2x per week resulting in greater muscle size than 4x, with too much confidence.

- It would be worth providing further context, or information around what other authors have done, regarding your comments on lines 257-258 about measuring changes in muscle mass.

- Ensure you clearly state that changes you discuss were non-significant when discussing them, e.g. lines 284-285, this will ensure readers are aware these changes were not statistically significant.

- As comparing training frequencies effects on explosive actions was a novel aspect stated in the introduction, I would have liked to see more coverage of the results of training frequency on explosive actions in your discussion - even though they were not significant.

Additional comments

Thank you for your efforts in conducting this study and producing the manuscript. I trust the comments are useful in enhancing your manuscript.

Reviewer 2 ·

Basic reporting

There are a number of grammar errors throughout especially in the Discussion. Tables and figures are clear, and appropriate references are cited.

Experimental design

Some methods are missing. Volume load data should be included.

Validity of the findings

My major concern is that these subjects were not very trained. The authors claim they have at least 2 years of experience but their 1RMs were quite low.

Additional comments

General Comments
Overall, the study addresses an interesting topic. My major comments are:
1. Some key Methods are missing.
2. Volume load data should be presented and analyzed.
3. The quality of the Discussion needs to greatly improve. There are a number of grammar errors throughout and better extrapolation of the results could be presented.
4. Although the authors’ claim these were trained men, their 1RM values are relatively low for a trained population. Thus, it may be questionable if these results can be applied to trained populations.

Specific Comments
Abstract, line 33: should mention these were resistance-trained subjects.
Line 55: Other important variables are intensity, exercise selection and sequence, and velocity.
Line 72: should be “Gentil.”
Line 83: the sentence “while less is known regarding muscle explosive actions performance and endurance in intermediately trained subjects” is awkward and should be restated.
Line 88: here again and throughout, “explosive actions performance” is awkward – why not just state “power performance.”
Line 106: here and throughout it should be “mass (kg)” not “weight.”
Line 112: this sentence is a run-on but how were subjects screened for supplement or drug use? Questionnaire? If so, did they quit prior to beginning or were they allowed to maintain use during study?
Line 125: should be “body mass.”
Line 128: no need to capitalize calipers. Should provide more detail. Were 2-3 skinfolds taken and averaged?
Line 140: more information should be provided for strength testing especially the leg press. How many trials? Rest intervals? How was range of motion standardized for this exercise? How was technique standardized for curl and BP?
Line 154: these are power measures, I suggest using the term “power” here and throughout. For Vertec, please provide the actual maker not the distributor. Power Systems is the distributor.
Line 136: what warm-up? Similar to previous warm-ups? More information is needed here.
Line 166: Technically the authors are referring to equated volume load, not volume. Volume just represents total reps. Volume load factors in loading. Thus, the term volume load should be used throughout.
Line 167: should be “…A 60 to 90 sec period of rest between sets…”
Line 168: I strongly recommend the authors include their volume load data for each group. It was stated no differences were observed between groups so the authors should include volume load data per week or at least first, middle, and last week plus the results of statistical analysis.
Line 191: should be “significant.” Same in line 199, 206, and 217.
Line 250-251: statement has grammar errors.
Discussion: there are a number of grammar errors throughout.

Reviewer 3 ·

Basic reporting

The manuscript presents clear professional English. The authors present expertise on this field.

Experimental design

The authors report the study as randomized-controlled design, to endorse this design and be noted as low-risk bias for further potential inclusion in systematic reviews and meta-analysis, the authors needs to improve the description of some domains that weaken the design.

The authors most describe how and Who did the randomization of participants (for exemple: by random sequence generator; a researcher who did not participated of study) (Page 4, line 91).

The inclusion and exclusion criteria most be cleary described before description of participants characteristics.

Need to be reported the reason why 5 participants were removed from study, as show in figure 2.

On page 5, line 105, most include a coma after "(RT2; n = 12".

On page 6, lines 122-123, the authors describes "All subjects were oriented to continue with their normal daily life activities and dietary intake throughout the study duration.", however, on page 6, lines 133-139, a nutritional specialist meet the participant each training session to asses their customary nutritional regime (i.e., approximately 25% protein, 25% fat and 50% carbohydrate). Please, clarify how did the authors unidentified this customary nutritional regime, there was a retrospective nutritional asses (at least six-month) previously the study?

The authors did not present any nutritional result in study. Please, clarify the absence of this result, since was recorded.

Validity of the findings

To improve validity of the findings, the authors needs to clarify the two-way analysis of variance with repeated measures (3 [group] x 3 [time]), for exemplo, the Levine test was performed to confirm the homogeneity of variances? If yes, assumed the homogeneity? If no, the Greenhouse-Geisser correction was applied?

Which statistical software was used to perform the analysis? Please, clarify these information.

On the results section, the authors needs simplify the description, for exemple, since present significantly interaction (group by time), there is not necessary describe the time and group effect separadely (the table shows from itself), it can confound the reader. Please, clarify this point.

On the discussion section (Page 10, lines 242-246) "Previous experimental studies reported two sessions per week RT induced small ES but in this study we found moderate increases in muscle size in arm, thigh and chest, suggesting benefits of two sessions per week RT for enhancing muscle size in comparison to RT4 (Gentil et al., 2015, Colquhoun et al., 2018; Zaroni et al., 2019; Yue et al., 2018).", why did the authors discuss the effect size found in experimental studies with the effect size in this current study (Human)? Please, clarify this point.

On page 10, lines 246-249, "The RT2 group performed 4 sets per exercise in each training session which may induce stimulation of muscular hypertrophy signalling pathways by increasing protein synthesis and providing mechanical stress in the muscle fibers in a greater magnitude than following the lower training volume during RT4 (Ogasawara et al., 2017).", Ogasawara et al. present that there is a relationship between high-volume resistance training and increase of protein synthesis, but the authors agree that the acute increase on protein synthesis promoted by resistance exercise may not reflect in chronic condition?

If suitable in this context, there is a recent paper (2019) regarding the chronic effect of resistance training on muscular hypertrophy signaling "Padilha CS, Cella PS, Ribeiro AS, et al. Moderate vs high-load resistance training on muscular adaptations in rats. Life Sci. 2019;238:116964.", if is not suitable for the discussion, please disregard.

Additional comments

It is a pleasure to read this study. The authors present several good and strong points regarding the frequencies with volume-equated. All points raise in this review addressed to the authors are entirely to improve, or at least try to, the scientific contribution in resistance training field.

---

## Round 0.2 · Minor Revisions

I thank the authors for attending to almost all of the comments of the three reviewers and me in their resubmission. Some small things that still require amendments include the following points. The line numbers I quote are with respect to the track changes word document.

Line 112: this should be spelt as physician.

Line 227 – 229: can you please provide more information here about how the maximum distance of the throw was recorded? For example, was there a floor-mounted tape measure that was used to record distance and was the distance estimated by sight or via a video camera recording? Further, what part of the ball was used with respect to measuring the distance on impact?

Table 1: the table heading should read “……at pre-and in the middle of the training period (mean ± SD)”.

Reviewer 2 ·

Basic reporting

The manuscript has improved in this area. I have no further comments.

Experimental design

The design is fine. I am glad the authors changed "resistance-trained" to "recreationally trained" given my previous comments questioning the strength level of these subjects.

Validity of the findings

No Comment

Additional comments

The manuscript has improved and I do not have any further edits. I am glad the authors changed subject status to "recreationally trained" as I feel it is more appropriate given their level of strength.

---

## Round 0.3 · accepted · Accept

The reviewers are satisfied that you have adequately addressed their concerns.